# Effect of Occupant Activity on Indoor Particle Concentrations in Korean Residential Buildings

**Hyungkeun Kim** [1] 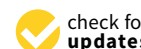**, Kyungmo Kang** [1,2] **and Taeyeon Kim** [1,*]

[1]   Department of Architectural Engineering, Yonsei University, Seoul 03722, Korea;
      hang0621@hanmail.net (H.K.); kyungmokang@kict.re.kr (K.K.)
[2]   Department of Living and Built Environment Research, Korea Institute of Construction Technology,
      Goyang-Si 10233, Korea
[*]   Correspondence: tkim@yonsei.ac.kr

**Abstract:** Due to the recent industrial development and COVID-19 pandemic, people are spending more time indoors. Therefore, indoor air quality is becoming more important for the health of occupants. Indoor fine particles are increased by outdoor air pollution and indoor occupant activities. In particular, smoking, cooking, cleaning, and ventilation are occupant activities that have the largest impact on indoor particle concentrations. In this study, indoor and outdoor particle concentrations were measured in ten apartment houses in South Korea for 24 h. Indoor particle concentrations were measured in the kitchen and living room to evaluate the impact of cooking, one of the most important sources of indoor particles. An occupant survey was also conducted to analyze the influence of occupant activities. It was found that the impact of outdoor particles on indoor particle concentrations in winter was not significant. The largest particle source was cooking. In particular, a large amount of particles was generated by broiling and frying. In addition, cooking-generated particles are rapidly dispersed to the living room, and this was more obvious for small particles. It is expected that this result will be statistically generalized if the particle concentration of more houses is analyzed in the future.

**Keywords:** indoor air quality; fine particle; occupant activity; cooking-generated particle; ventilation; range hood

---

## 1. Introduction

Today, people are spending most of their time indoors [1]. In particular, telecommuting has increased due to the COVID-19 pandemic of late, and this trend is expected to continue for a while [2]. Therefore, the indoor living environment is becoming more important. In particular, the importance of indoor air quality, which may have a significant influence on the health of occupants, is attracting more attention [3]. If occupants are exposed to poor IAQ(Indoor Air Quality), they may get respiratory and skin diseases [4].

In indoor living space, many pollutants are generated by infiltration from the outside, ventilation, building materials, and occupant activities [5]. There are various indoor pollutants, including carbon dioxide, formaldehyde, volatile organic compounds, and fine particles [6–9]. Among them, fine particles cause serious respiratory and cardiovascular diseases when introduced into the body through the respiratory system. It was reported that carcinogenic potential may increase if people are continuously exposed to fine particles [10].

The main cause of indoor particles is the penetration of outdoor particles into the inside through ventilation or infiltration [11]. The houses that have been recently built, however, have improved airtightness [12]. Therefore, when windows are closed, the influence of outdoor particles on indoor

particle concentrations is relatively small. The causes of indoor particles include cleaning, smoking, and cooking [13,14]. Among them, smoking and cooking are indoor activities that have the largest impact on indoor particle concentrations [15–17]. Cooking is reported to be the biggest cause of lung cancer among housewives who usually cook in the kitchen [15–19]. Smoking and cooking generate particles of very small size (e.g., size of 1 μm or less) due to combustion [20]. Depending on the particle size, the part deposited in the body is different [21]. Small-sized particles cannot be filtered by the respiratory tract and can penetrate into the lungs and be deposited in the alveolar region [22]. The particles having a size under 0.4μm are mostly deposited in the alveolar region [23].

Cooking is the most important particle source in non-smoking houses [18,19]. In the cooking process, in particular, cooking-generated particles can be rapidly dispersed to adjacent spaces due to the generation of high heat [24]. Particles in the kitchen can be rapidly removed by operating the range hood [25–27]. The particles dispersed to other spaces, however, stay longer indoors because it is difficult to discharge them through the range hood, and they may have an adverse effect on occupants. Ventilation is necessary to effectively discharge indoor fine particles. In particular, when the outdoor particle concentration is low, fresh air can be supplied through natural ventilation, and indoor particle concentration can be rapidly decreased. Reduced indoor particle concentration can reduce the health risk of occupants [28]. In addition, as indoor particle concentration decreases, the mental illness of occupants may decrease [29].

There have been many studies on indoor particle concentration and occupant activities. Bhangar et al. analyzed particle source by measuring the concentration of ultrafine particles in seven houses in the United States. It was found that indoor fine particles are greatly affected by indoor activities and infiltration from outdoors [30]. Abt et al. analyzed the concentration and size of fine particles indoors and outdoors of four houses in Boston, USA. The analysis concluded that indoor fine particle concentration is affected in combination with indoor and outdoor sources [31]. As such, the indoor particle concentration is affected by indoor activities as well as outdoor particle concentration. Although there have been many studies on the causes of indoor particles, studies on indoor particle concentrations in apartment houses, their causes, and dispersion of particles in indoor space are still not sufficient. For the influence of indoor particles in apartment houses, the schedules and activities of occupants as well as the shape of space must be considered. Therefore, it is necessary to classify various indoor elements and quantitatively identify their influence on the generation and dispersion of indoor particles.

Indoor occupant activities have a close correlation with indoor particle concentrations in apartment houses. Indoor particle concentrations may vary depending on various parameters, such as the ventilation method of the building, occupant activities, and building conditions. Therefore, it is necessary to analyze occupant activities through occupant surveys under various conditions. The impact of these activities on indoor particle concentration should also be analyzed. To analyze this, indoor and outdoor particle concentrations were analyzed for ten apartment houses located in downtown areas in South Korea. In addition, the schedules and indoor activities of occupants were investigated through an occupant survey that had not been conducted before. Through the analysis of these data, the causes of indoor particles and the effect of occupant activities on the generation and dispersion of indoor particles were quantitatively evaluated.

## 2. Materials and Methods

### 2.1. Description of Target Buildings

In this study, the effect of occupant activities on the concentrations and distribution of indoor particles was evaluated. To this end, particle concentrations were measured and an occupant survey was conducted on ten apartment houses located in downtown areas in South Korea. According to the Korean APT House Living Condition Statistics, more than 80 percent of buildings are medium-sized apartments of 59 to 140 square meters [32]. Most of the medium-sized apartment houses in Korea

have an open kitchen that is adjacent to the living rooms of the house [33]. The target buildings were selected by residents of medium-sized apartment houses in urban areas with the intent to participate. Five houses are located in the Seoul-Gyunggi province, and the other five are located in Daejeon (Figure 1). Table 1 shows the information on the ten apartment houses selected. For the target buildings, buildings in which two or more people were living were selected. Cooking activities may have an important influence on indoor particles [34,35]. Therefore, the fuel type and range hood operation were investigated to evaluate the generation of indoor particles during cooking. In addition, the cooking type (broiling, frying, and soup) was investigated, and meal photographs were requested to analyze the difference depending on the cooking method. In all the target buildings, range hoods were operated, and electricity was used as fuel in two houses. For ventilation, natural ventilation through windows was performed in all the houses. Measurements were conducted in winter. Winter in South Korea has very low ambient air temperatures (average: −0.58 °C) and relatively high fine particle levels. Therefore, the occupants did not perform cross-ventilation by opening all windows. Instead, single-sided ventilation was performed in all houses. And to remove odors or pollutants from cooking, occupants turned on range hoods. When operating the range hood, the window was opened simultaneously to supply the make-up air of the range hood.

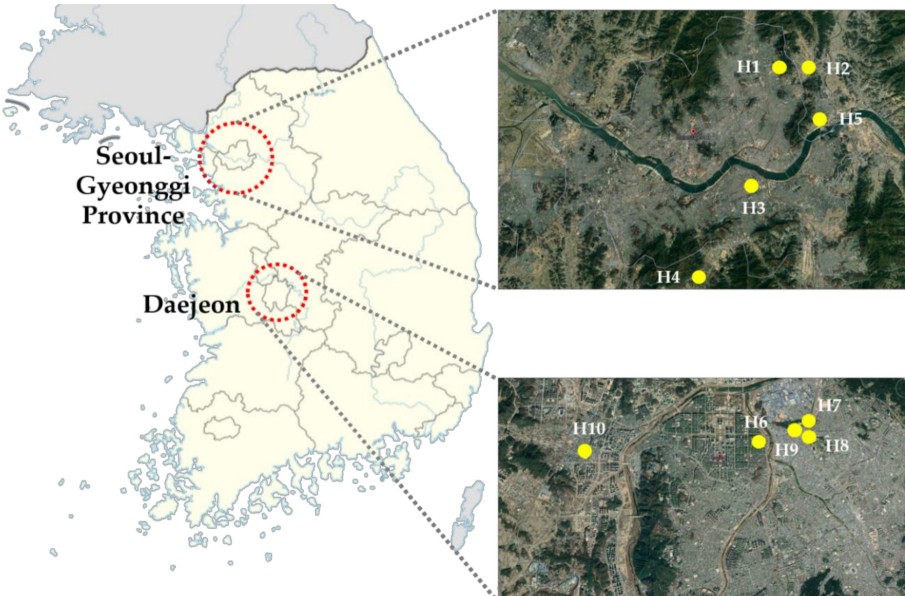

**Figure 1.** Measurement locations of 10 houses.

**Table 1.** Description of target buildings.

| Building No. | Floor Areas (M²) | Floor Height (1) (#) | Built Year | No. of Occupants | Type of Fuels | Type of Ventilation (2) |
|---|---|---|---|---|---|---|
| H1 | 64.0 | 5/5 | 1975 | 2 | LNG | H, N.V |
| H2 | 164.0 | 21/25 | 2016 | 3 | LNG | H, N.V |
| H3 | 202.0 | 23/29 | 1999 | 4 | Electricity | H, N.V |
| H4 | 84.9 | 14/20 | 2001 | 4 | Electricity | H, N.V |
| H5 | 77.7 | 3/20 | 2006 | 4 | LNG | H, N.V |
| H6 | 59.0 | 10/15 | 2001 | 2 | LNG | H, N.V |
| H7 | 130.0 | 13/15 | 1991 | 3 | LNG | H, N.V |
| H8 | 122.7 | 13/15 | 1991 | 2 | LNG | H, N.V |
| H9 | 130.0 | 12/15 | 1992 | 2 | LNG | H, N.V |
| H10 | 138.8 | 7/15 | 2016 | 4 | LNG | H, N.V |

(1) Floor Height—floor of target housing unit/floor of building (e.g., 21/25 is the 21st floor of a 25-story building).
(2) Type of Ventilation—H: range hood; N.V: natural ventilation.

## 2.2. Field Measurement

Field measurements for the ten houses were performed from December to January in winter. The field measurements were performed for 24 h, and indoor and outdoor particle concentrations (particles with diameters that were 10 μm and smaller) were measured at 30-s intervals. Indoor and outdoor particle number concentrations were measured to evaluate the influence of indoor and outdoor particles. Particle concentrations were measured using an optical particle counter (OPC; TSI, Model 3330, USA). This measuring instrument counts the particle number for each channel using the light scattering method. In this study, measurement was performed in six particle size ranges (0.3–0.5, 0.5–0.7, 0.7–1.0, 1.0–2.5, 2.5–5.0, and 5.0-10.0 μm). In addition, indoor temperature and humidity were measured using a digital IAQ meter (Testo, Model 480, Titisee-Neustadt, Germany).

Measurement was performed for the kitchen, living room, and outdoor air. In the kitchen, the measuring equipment was located around the cookstove to evaluate the influence of cooking-generated particles. In the living room, the equipment was located in the main activity area of occupants. For outdoor particle concentrations, the equipment was located by a window in the balcony with the sampling tube exposed to the outside and taped tightly to prevent infiltration. Figure 2 shows the installation locations of the measuring equipment.

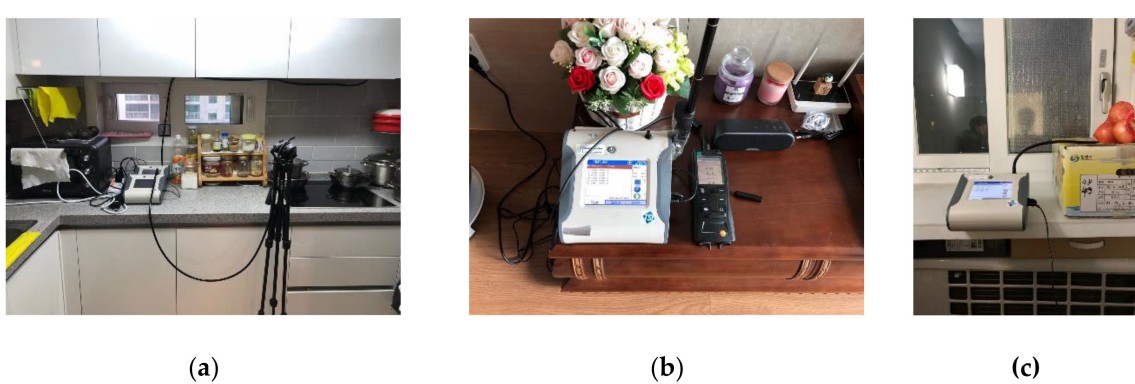

|  (**a**)  |  (**b**)  |  (**c**)  |

**Figure 2.** Measurement location for particle concentration: (**a**) Concentration of the kitchen (**b**) Concentration of the living room; (**c**) Outdoor particle concentration.

## 2.3. Occupant Survey

Occupant activities may have a significant influence on indoor particle concentrations [36]. To evaluate the effect of occupant activities on indoor particle concentrations, it is necessary to identify the detailed schedules and indoor activities of occupants. In this study, an occupant survey was conducted to collect information on occupant activities. The time spent on cleaning, smoking, ventilation, and cooking activities was surveyed. For cooking, in particular, the start time, end time, cuisine, and range hood operation were surveyed (Figure 3).

## 2.4. I/O Ratio and L/K Ratio

The I/O ratio is an indicator that shows the relationship between indoor and outdoor particle concentrations. It is easy to understand and has been widely used in the field of indoor air quality because it directly shows the influence of outdoor air on indoor particles [37]. The I/O ratio is expressed as the ratio of the indoor particle concentration to the outdoor particle concentration as follows:

$$\text{I/O ratio} = \frac{C_{in}}{C_{out}} \tag{1}$$

where $C_{in}$ is indoor particle concentration, and $C_{out}$ is outdoor particle concentration.

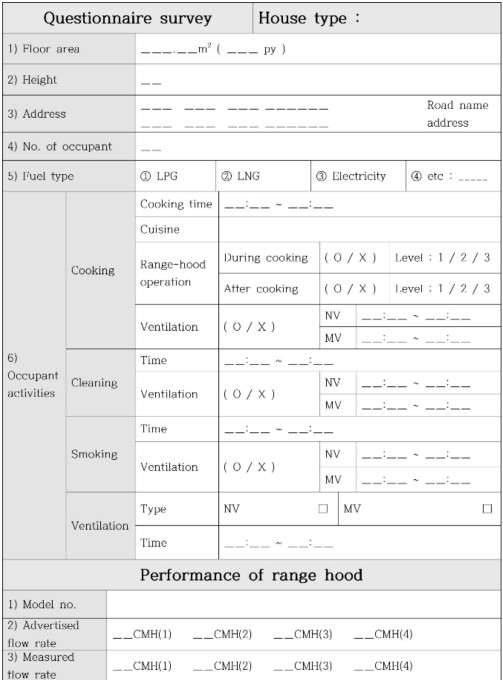

**Figure 3.** Questionnaire survey of occupants' activities.

Colbeck et al. analyzed the indoor and outdoor particle concentration at three homes in one urban and two rural areas of Pakistan. As a result, the average I/O ratio of urban area houses was 1.71 due to high indoor activity [38]. Jones et al. measured the particle concentration of seven houses in Birmingham, England. The I/O ratio of houses in urban areas was 1.0–3.9 on average, while those on the roadside had a smaller I/O ratio due to the influence of the outdoor air contaminants [39]. Matson analyzed the city I/O ratio in a Scandinavian urban area (5 office and 3 residential buildings), and the range of the I/O ratio was about 0.5–0.8 [40]. Mönkkönen et al. analyzed the number concentration of indoor and outdoor fine particles of two houses by season in Nagpur, a city located west of India. The results showed that the colder the weather, the smaller the I/O ratio [41].

The L/K ratio represents the ratio of the particle concentration of the living room to that of the kitchen [42]. As mentioned earlier, cooking in the kitchen generates a high concentration of fine particles. Cooking-generated particles can be rapidly removed by range hood operation, but some of the particles that cannot be discharged may be dispersed to the living room. As the dispersion of the fine particles varies over time, the L/K ratio can be expressed as a time series [43]. Usually, the concentration of the kitchen and living room is about the same. However, when occupants cook in the kitchen, the particle concentration in the kitchen increases, and the L/K ratio decreases. A high L/K ratio on cooking conditions indicates that cooking-generated particles are rapidly dispersed to the living room [44,45].

## 3. Results and Discussion

### 3.1. Result of Occupant Activities

First, natural ventilation through windows was performed in all of the ten houses. Natural ventilation was performed for 23 min on average. In South Korea, the air temperature and relative humidity are low, and outdoor particle concentrations are high in winter. As the measurement period was in December, the ventilation time was not relatively long. Cleaning was performed once for 24 h in six houses. The cleaning time ranged from 15 to 30 min.

Table 2 shows the result of occupants' activities and airflow rate of the range hood. It was found that cooking was performed twice or more in all the houses. Range hoods were mostly operated during

cooking and turned off just after cooking. There was no house in which the range hood was operated after cooking. Three cooking methods were investigated. Among the cooking methods, broiling was performed 15 times, frying 10 times, and soup (boiling) 16 times. Soup and broiling methods were performed more frequently. The measured range hood airflow rate was between 81 and 308 CMH. In most of the houses, range hoods were operated at a low airflow rate due to noise. During the operation of the range hood, make-up air was supplied through windows in all houses. Therefore, it was assumed that the airflow rate of natural ventilation was the same as the exhaust airflow rate of the range hood.

**Table 2.** Result of occupants' activities and airflow rate of the range hood.

| | Ventilation (n [1]) | Cleaning (n) | Cooking Event [2] | Cooking Type [3] | Airflow Rate of Range Hood (CMH [4]) |
|---|---|---|---|---|---|
| H1 | - | - | Bf/D | S/B | 285 |
| H2 | 2 | - | Bf/L/D | F/S/S | 81 |
| H3 | 2 | 1 | L/D | F/F,S | 150 |
| H4 | 4 | 1 | Bf/L/D | S/B/B,F,S | 207 |
| H5 | 4 | 1 | Bf/L/D | F,S/B/B,S | 119 |
| H6 | 3 | - | Bf/L/D | S/S/B,S | 245 |
| H7 | 2 | 1 | Bf/L/D | S/ B,F,S/B | 95 |
| H8 | 4 | 1 | Bf/L/D | F,S/ B,F,S/B | 95 |
| H9 | 4 | 1 | Bf/L/D | B/B/B,F | 125 |
| H10 | 1 | - | Bf/D | B,S/B,F,S | 308 |

[1] n: number of times; [2] Cooking event: Bf/L/D—breakfast/lunch/dinner; [3] Type of cooking: B/F/S—broiling/frying/soup; [4] CMH: m$^3$/h.

### 3.2. Result of Indoor and Outdoor Particle Concentration

Figure 4 shows the measured indoor and outdoor particle concentrations. For most of the houses, the outdoor particle concentration was found to be higher than the indoor particle concentration. In particular, H5 located near a railway station and H10 located near a wide road exhibited high outdoor particle concentrations. Moreover, the outdoor concentration was found to be relatively higher than the indoor concentration for most of the buildings. The influence of the outdoor concentration on the indoor concentration can be simply evaluated through the I/O ratio. The I/O ratio was mostly less than 1, indicating that the influence of outdoor particles was not significant. For six houses, in particular, the I/O ratio was less than 0.5, showing that the influence of outdoor air was very small. In the case of H7 with an I/O ratio of more than 1, the indoor concentration appears to have been increased by indoor occupant activities because the outdoor concentration was low.

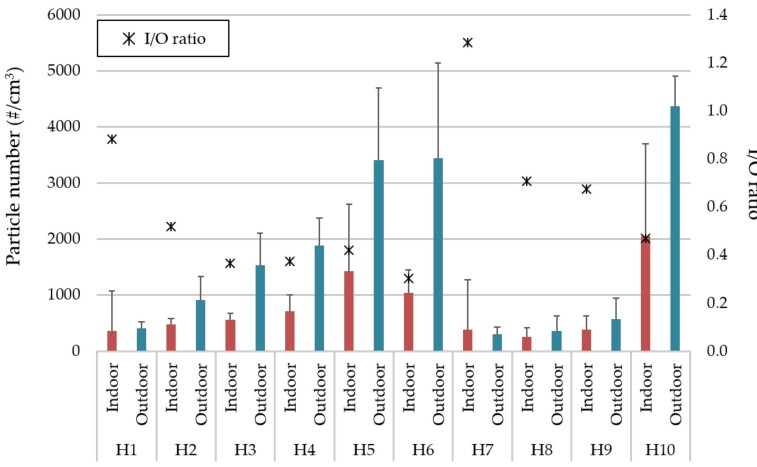

**Figure 4.** Indoor and outdoor particle concentration and the I/O ratio.

Table 3 shows the average I/O ratio and the frequency of ventilation and cooking. In all houses except H10, the I/O ratio increased in ventilation compared to the average I/O ratio. When outdoor particle concentration is high, indoor particle concentration may increase by natural ventilation. If the I/O ratio is greater than 1, it indicates that a large amount of fine particles is generated indoors. The I/O ratio was greater than 1 when occupants ventilated in H2, H4, H7, and H8. As mentioned in chapter 2, occupants ventilated by opening windows during cooking. Therefore, it appears that the indoor particle generation is greater than particle penetration from outdoors. During cooking, the I/O ratio increased significantly compared to the 24-h average I/O ratio. This is because a large amount of particles was generated during cooking. In addition to not many samples, there was also no clear relationship between the ventilation frequency and the I/O ratio. This is because the ventilation frequency was low, and the time of window opening was as short as less than ten minutes even if ventilation was performed, and thus the influence of outdoor particles on indoor particles was not significant. It appears that the influence of outdoor air due to infiltration was not significant because the recently built apartment houses had excellent airtightness.

**Table 3.** Ventilation frequency and the I/O ratio.

| Case | H1 | H2 | H3 | H4 | H5 | H6 | H7 | H8 | H9 | H10 |
|---|---|---|---|---|---|---|---|---|---|---|
| Ventilation frequency | 0 | 2 | 2 | 4 | 4 | 3 | 2 | 4 | 4 | 1 |
| Cooking frequency | 2 | 2 | 1 | 3 | 2 | 2 | 2 | 3 | 2 | 2 |
| I/O ratio (24 h average) | 0.88 | 0.52 | 0.37 | 0.37 | 0.42 | 0.30 | 1.28 | 0.71 | 0.67 | 0.47 |
| I/O ratio (Ventilation) | - | 1.38 | 0.58 | 1.50 | 0.88 | 0.48 | 5.04 | 1.08 | 0.70 | 0.38 |
| I/O ratio (Cooking) | 6.37 | 1.67 | 0.66 | 2.63 | 2.24 | 0.66 | 8.79 | 2.20 | 0.78 | 1.92 |

Figure 5 shows the particle size distribution of indoor and outdoor air. Particles in the 0.3–0.5 μm size range exhibited the largest proportion for both indoor and outdoor air. In particular, particles smaller than 1 μm, which can be seen as small particle size, accounted for more than 95% in indoor and outdoor air in all cases. Small particles can be deposited in the respiratory system once introduced into the body [21]. In this case, the particles may adversely affect human health due to their deposition on the alveolar of the lungs. Therefore, it is necessary to manage the absolute indoor particle concentration. In the case of indoor particles, particles in the 1–10 μm size range exhibited somewhat high proportions in most of the houses.

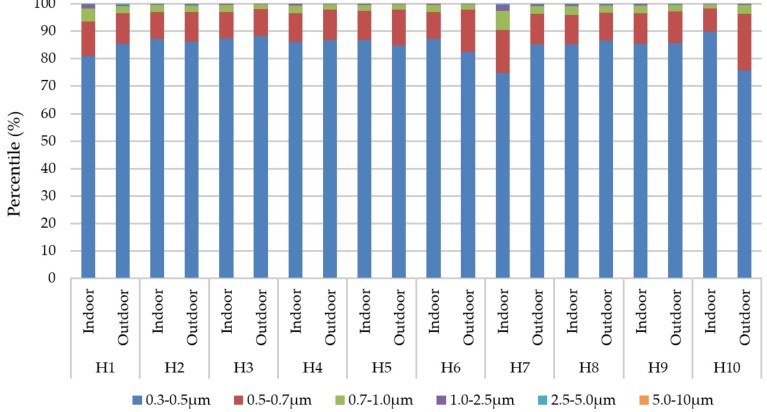

**Figure 5.** Particle size distribution of indoor and outdoor air.

### 3.3. Result of Indoor Particle Concentration and Occupants' Activities

Figure 6 shows the particle concentrations in the living room and kitchen as well as occupant activities. In the figure, H4 and H5, which had the most indoor activities, were compared. The measurement data of the remaining houses are included in the Appendix A section. First, except when occupants cooked, indoor particle concentration was lower than outdoor particle concentration for most of the day. When ventilation was performed by opening windows, the indoor particle concentration tended to be high if the outdoor particle concentration was high in some houses, but there was no significant influence in most cases. This appears to be because the time of living with windows closed was long due to the low ventilation frequency. In most cases, the largest amount of indoor particles was generated during cooking. Table 4 shows the maximum indoor particle concentration and concentration difference (ΔC) for each activity. ΔC represents the difference between the starting concentration and the maximum concentration of each activity. As described earlier, indoor particle concentration increases the most during cooking in all cases. Furthermore, the maximum concentration is highest. During the measurement period, cleaning was performed in six houses, but there was no significant increase in indoor particle concentration before and after cleaning. This appears to be because the latest high-performance vacuum cleaners were used in most houses and ventilation was performed at the same time by opening windows under low outdoor air conditions in some houses.

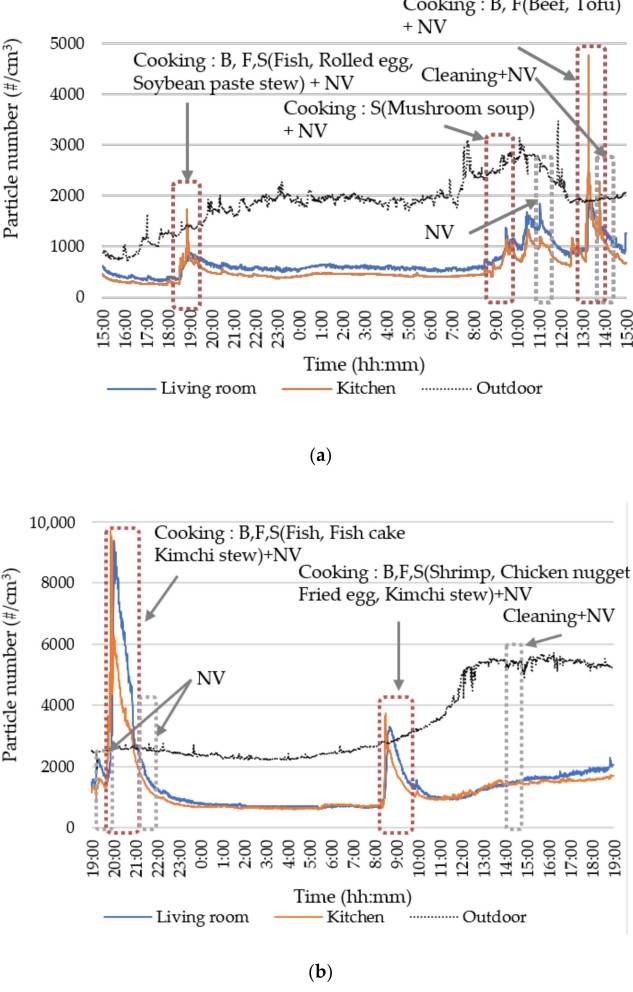

**Figure 6.** Particle concentration of the kitchen and living room and occupant activities: (**a**) Particle concentration of H4; (**b**) Particle concentration of H5.

**Table 4.** Maximum particle concentration and $\Delta C_{in}$ for occupant activities ($\times 10^3$ particles/cm$^3$).

| Activities | | H1 | H2 | H3 | H4 | H5 | H6 | H7 | H8 | H9 | H10 |
|---|---|---|---|---|---|---|---|---|---|---|---|
| Cooking | Max | 8.25 | 2.52 | 1.93 | 4.77 | 9.38 | 1.82 | 13.40 | 8.50 | 9.58 | 11.14 |
| | $\Delta C_{in}$ | 8.09 | 2.15 | 1.61 | 4.07 | 7.87 | 1.01 | 13.21 | 8.49 | 8.83 | 10.05 |
| Ventilation | Max | - | 1.89 | 0.91 | 1.85 | 1.59 | 0.76 | 0.77 | 0.69 | 1.13 | 1.75 |
| | $\Delta C_{in}$ | - | 1.52 | 0.55 | 0.52 | 0.27 | 0.41 | 0.56 | 0.11 | 0.34 | 0.48 |
| Cleaning | Max | - | - | 0.91 | 2.28 | 1.56 | - | 1.40 | 0.69 | 1.13 | - |
| | $\Delta C_{in}$ | - | - | 0.55 | 1.06 | 0.17 | - | 0.20 | 0.19 | 0.34 | - |

Table 5 shows the particle number concentrations by particle size in the living room and kitchen for the entire measurement period and for one hour during cooking. The measurement data of the outdoor concentrations are included in Appendix B. In general, the particle concentration was high in the living room because the main space for occupant activities is the living room. In addition, the L/K ratio was more than 1 during cooking in half the cases, indicating that particles are also likely to be dispersed to the living room during cooking. In particular, the L/K ratio was relatively higher for smaller particles. This confirmed that smaller particles are more likely to be dispersed to the living room. Therefore, cooking may adversely affect the health of occupants in other spaces if there are no proper countermeasures.

As shown in Figure 6, a large amount of particles was generated particularly during broiling and frying. During cooking, range hoods were operated in all the houses. In most cases, ventilation was performed by opening windows while range hoods were in operation. Nevertheless, the indoor particle concentration increased. In addition, the particle concentration in the living room sharply increased during cooking. In some houses, in particular, the particle concentration was higher in the living room than in the kitchen. This indicates that particles are rapidly dispersed to adjacent spaces despite the operation of the range hood. This tendency was generally more obvious in small houses.

Several studies have analyzed the number concentration of fine particles generated during cooking in the residential buildings. In the study of Wallace et al., the particle number concentration was around $1.3 \times 10^4$/cm$^3$ for dinner and $5.7 \times 10^3$/cm$^3$ for breakfast [46]. He et al. measured the concentration of fine particles under cooking conditions in Australian houses. The concentration in the kitchen was about $2.86 \times 10^4$/cm$^3$ [47]. See and Balasubramanian measured the concentration of fine particles in the kitchen during cooking in Singapore, which has a housing form similar to that of Korea. The concentration of fine particles varied depending on the cooking method, which was around $5.459 \times 10^4$/cm$^3$ [48]. Judging from the results of previous studies and measurements of this study, the concentration of fine particles increases significantly depending on the method of cooking.

**Table 5.** Particle concentration of the kitchen and living room.

| Case | | H1 | | H2 | | H3 | | H4 | | H5 | | H6 | | H7 | | H8 | | H9 | | H10 | |
|---|---|---|---|---|---|---|---|---|---|---|---|---|---|---|---|---|---|---|---|---|---|
| Particle Size (µm) | | L [1] | K [2] | L | K | L | K | L | K | L | K | L | K | L | K | L | K | L | K | L | K |
| 24 h | 0.3–0.5 | 288.96 | 252.36 | 412.44 | 335.31 | 489.17 | 373.75 | 607.91 | 472.60 | 1242.20 | 1092.15 | 906.84 | 620.70 | 290.47 | 274.70 | 213.99 | 195.69 | 325.51 | 380.19 | 1838.34 | 1366.45 |
| | 0.5–0.7 | 45.13 | 46.61 | 45.97 | 40.90 | 52.75 | 46.27 | 73.10 | 68.66 | 150.29 | 132.11 | 100.80 | 66.70 | 60.69 | 72.03 | 26.52 | 30.79 | 41.62 | 63.38 | 173.86 | 138.52 |
| | 0.7–1.0 | 17.26 | 16.90 | 12.29 | 10.33 | 14.27 | 12.60 | 20.56 | 20.21 | 32.94 | 23.03 | 26.95 | 16.58 | 27.10 | 31.62 | 8.47 | 9.42 | 11.36 | 16.28 | 32.72 | 18.44 |
| | 1.0–2.5 | 5.01 | 5.35 | 1.94 | 1.61 | 2.52 | 2.61 | 3.85 | 3.93 | 4.06 | 3.47 | 3.62 | 2.29 | 8.41 | 12.90 | 1.62 | 2.48 | 1.74 | 4.79 | 3.20 | 2.34 |
| | 2.5–5.0 | 0.88 | 0.91 | 0.45 | 0.35 | 0.36 | 0.38 | 0.38 | 0.39 | 0.46 | 0.41 | 0.30 | 0.20 | 1.16 | 1.73 | 0.26 | 0.59 | 0.34 | 1.01 | 0.29 | 0.22 |
| | 5.0–10 | 0.10 | 0.13 | 0.06 | 0.05 | 0.03 | 0.04 | 0.04 | 0.05 | 0.05 | 0.05 | 0.04 | 0.04 | 0.11 | 0.27 | 0.03 | 0.11 | 0.04 | 0.19 | 0.03 | 0.03 |
| | Total | 357.34 | 322.25 | 473.14 | 388.55 | 559.10 | 435.65 | 705.83 | 565.84 | 1430.00 | 1251.23 | 1038.55 | 706.51 | 387.94 | 393.25 | 250.89 | 239.08 | 380.61 | 465.84 | 2048.43 | 1526.01 |
| Cooking condition | 0.3–0.5 | 2491.07 | 2241.16 | 628.58 | 657.53 | 513.62 | 579.64 | 1047.78 | 1001.54 | 4805.04 | 3730.20 | 933.34 | 732.73 | 1963.12 | 2384.56 | 248.37 | 414.54 | 372.07 | 1416.35 | 6416.54 | 4406.23 |
| | 0.5–0.7 | 432.82 | 489.30 | 74.89 | 79.04 | 90.21 | 152.95 | 233.40 | 303.50 | 780.93 | 620.16 | 203.65 | 169.46 | 812.20 | 1025.22 | 31.37 | 67.55 | 44.36 | 307.89 | 620.98 | 423.47 |
| | 0.7–1.0 | 195.83 | 211.90 | 19.39 | 17.95 | 38.01 | 71.07 | 108.60 | 141.77 | 189.94 | 126.03 | 105.22 | 79.02 | 444.50 | 510.60 | 9.91 | 18.51 | 12.15 | 92.05 | 122.70 | 64.85 |
| | 1.0–2.5 | 76.09 | 86.78 | 3.34 | 2.95 | 12.36 | 23.74 | 34.85 | 44.40 | 27.16 | 25.12 | 19.20 | 11.54 | 160.25 | 240.53 | 31.12 | 66.30 | 1.58 | 36.20 | 18.86 | 14.47 |
| | 2.5–5.0 | 14.95 | 16.50 | 1.48 | 1.08 | 1.86 | 3.76 | 3.86 | 5.23 | 4.12 | 3.91 | 0.49 | 0.31 | 21.90 | 34.80 | 0.38 | 0.98 | 0.36 | 9.50 | 3.15 | 2.41 |
| | 5.0–10 | 1.95 | 2.44 | 0.27 | 0.21 | 0.16 | 0.53 | 0.26 | 0.63 | 0.51 | 0.59 | 0.10 | 0.08 | 1.90 | 5.54 | 0.04 | 0.17 | 0.04 | 2.06 | 0.40 | 0.34 |
| | Total | 3212.71 | 3048.08 | 727.94 | 758.75 | 656.22 | 831.68 | 1428.75 | 1497.08 | 5807.70 | 4506.01 | 1261.99 | 993.15 | 3403.87 | 4201.26 | 321.19 | 568.05 | 430.55 | 1864.06 | 7182.62 | 4911.77 |

[1] L: living room, [2] K: kitchen.

## 4. Conclusions

In this study, the effects of outdoor particles and occupant activities on indoor particle concentrations were evaluated for apartment houses in South Korea. Measurement was performed in winter when the outdoor particle concentration is relatively high. The results are as follows:

- The inflow of outdoor particles is not significant because the recently built apartment houses have excellent airtightness, and the ventilation frequency is not high in winter.
- Cooking has the greatest impact on indoor fine particle concentrations, and the impact of ventilation and cleaning is relatively small.
- In particular, a large amount of particles is generated during broiling or frying.
- Cooking-generated particles are rapidly dispersed to the living room. This tendency is more obvious for small particles.

This study had some limitations even though field measurements were performed for many houses. First, the time and duration of each occupant activity were requested during the survey, but there were many missing results. This made a more detailed analysis difficult. Second, the outdoor particle concentration was significantly different for each case because its measurement in the same period was difficult. In addition, better results could have been derived if measurement had been performed during a longer period instead of the 24-h measurement. Long-term measurement, however, was difficult because the measurement was performed for ordinary people. In the future, more practical results can be derived if more detailed analyses are conducted by long-term measurements over a week. Finally, more samples are necessary for the results of this study to be statistically generalized. Ten apartments were analyzed, but the location of the houses was concentrated in a specific area. These limitations may affect the generalization of the results of this study. The results can be statistically generalized if the concentration of fine particles in more houses is analyzed with the measurement data in this study.

**Author Contributions:** Conceptualization, methodology, field measurement, validation, and formal analysis, H.K. and K.K.; writing—original draft preparation, writing—review and editing, and visualization, H.K.; supervision, project administration, and funding acquisition, T.K. All authors have read and agreed to the published version of the manuscript.

**Funding:** This work was supported by the National Research Foundation of Korea (NRF) grant funded by the Korea government (MSIT, MOE) and (No. 2019M3E7A1113095).

**Acknowledgments:** This research was supported by the Basic Science Research Program through the National Research Foundation of Korea (NRF) funded by the Ministry of Science, ICT and Future Planning (NRF-2019R1I1A1A01062777).

**Conflicts of Interest:** The authors declare no conflict of interest.

## Appendix A. Particle Concentration of the Kitchen and Living Room and Occupant Activities

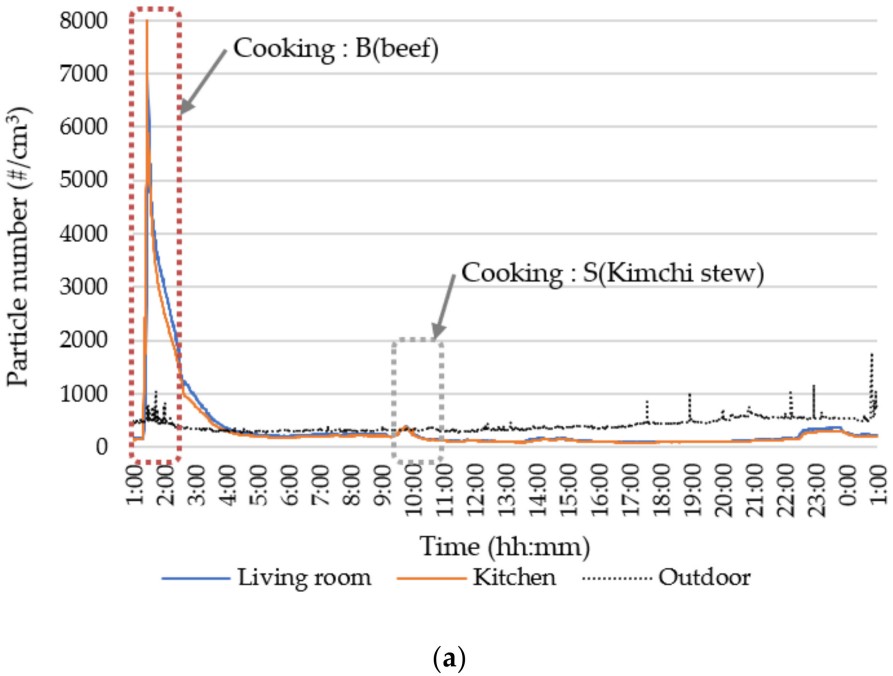

(**a**)

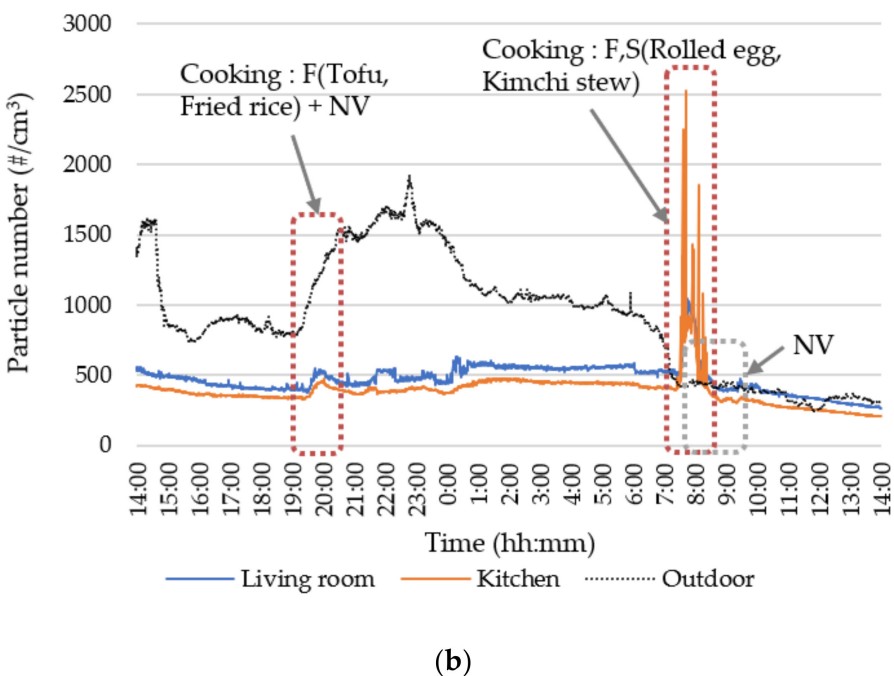

(**b**)

**Figure A1.** *Cont.*

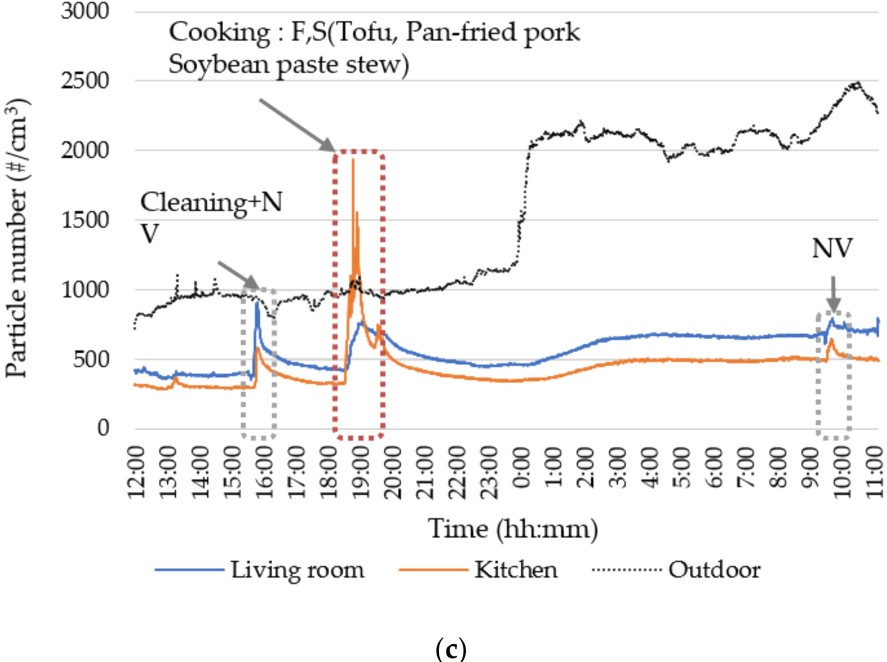

(**c**)

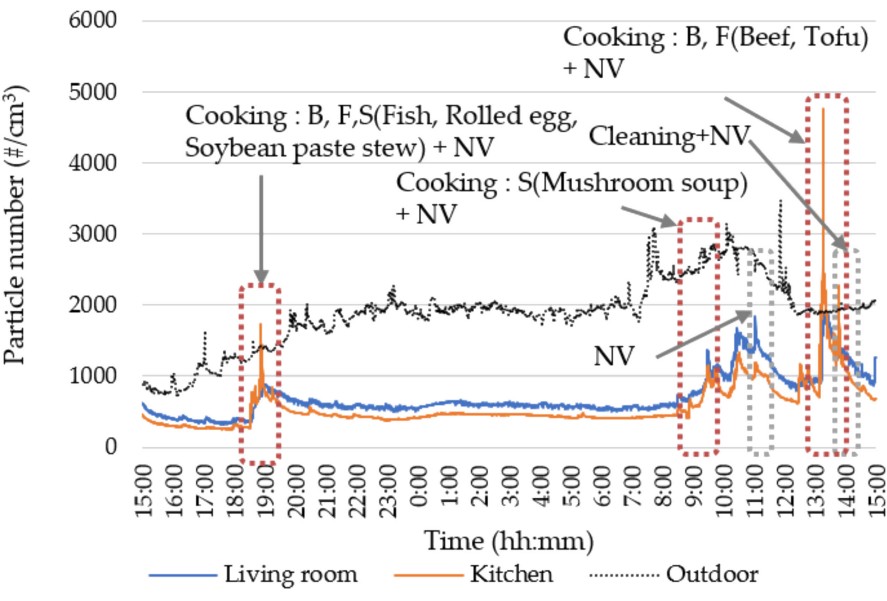

(**d**)

**Figure A1.** *Cont*.

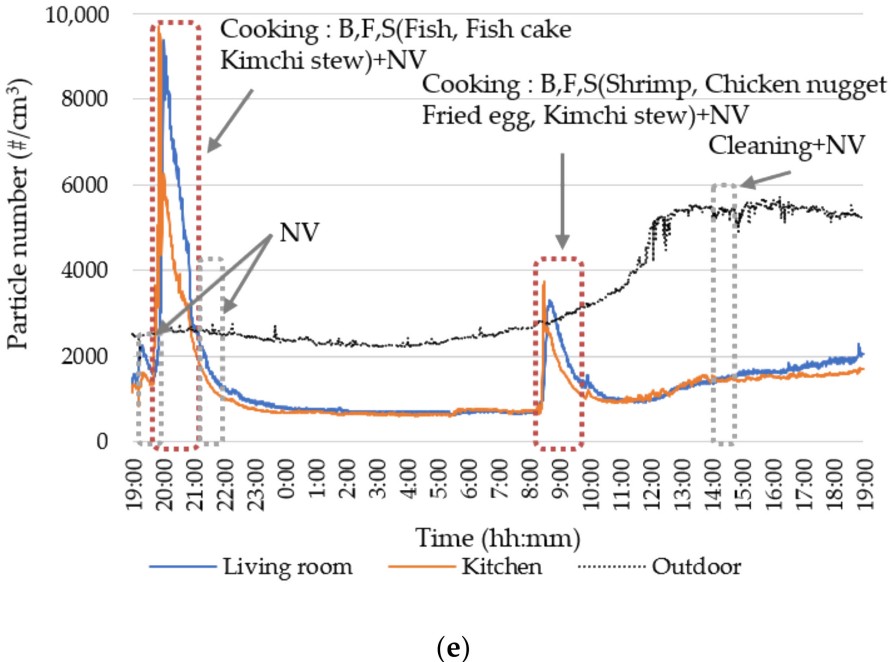

(**e**)

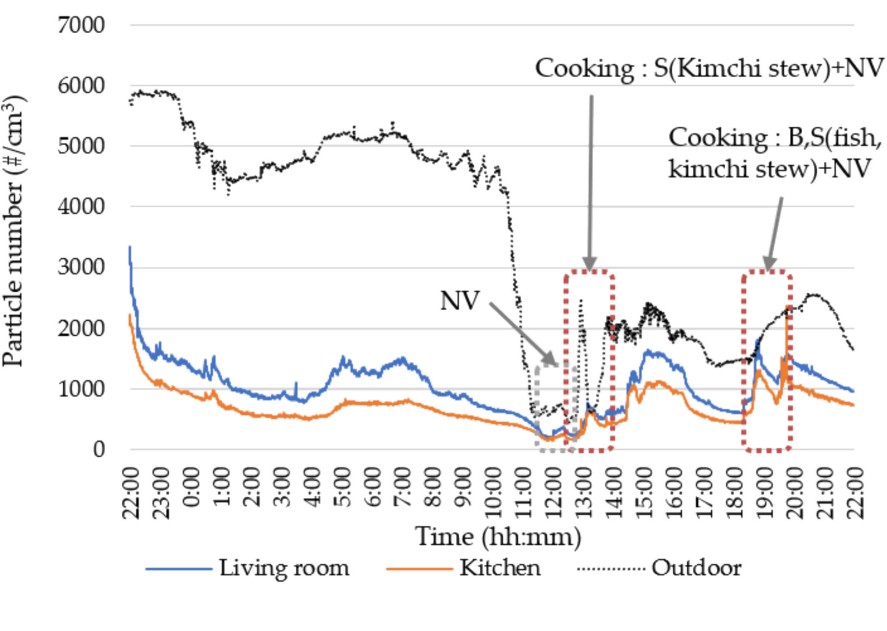

(**f**)

**Figure A1.** *Cont*.

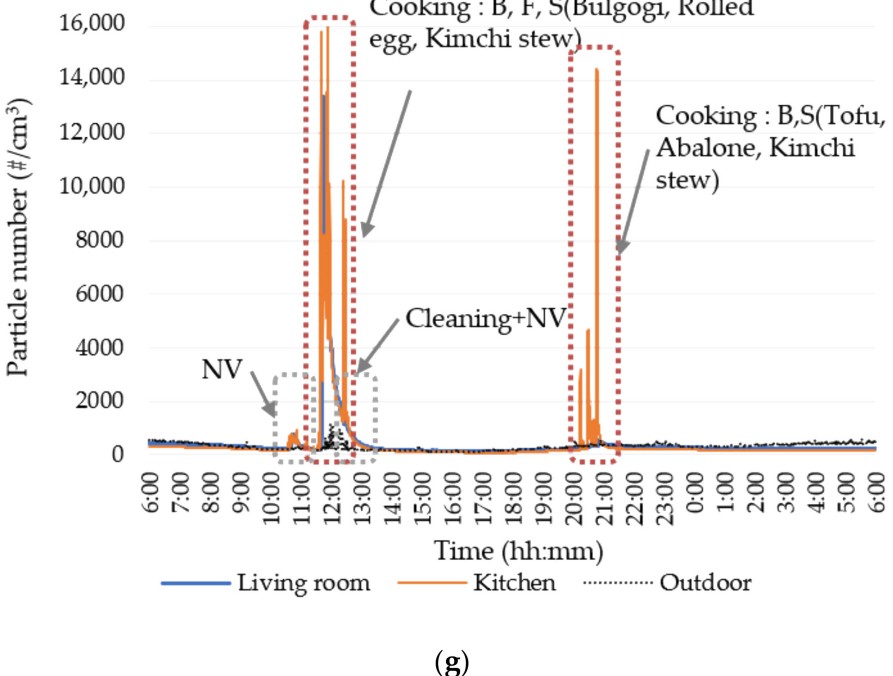

(**g**)

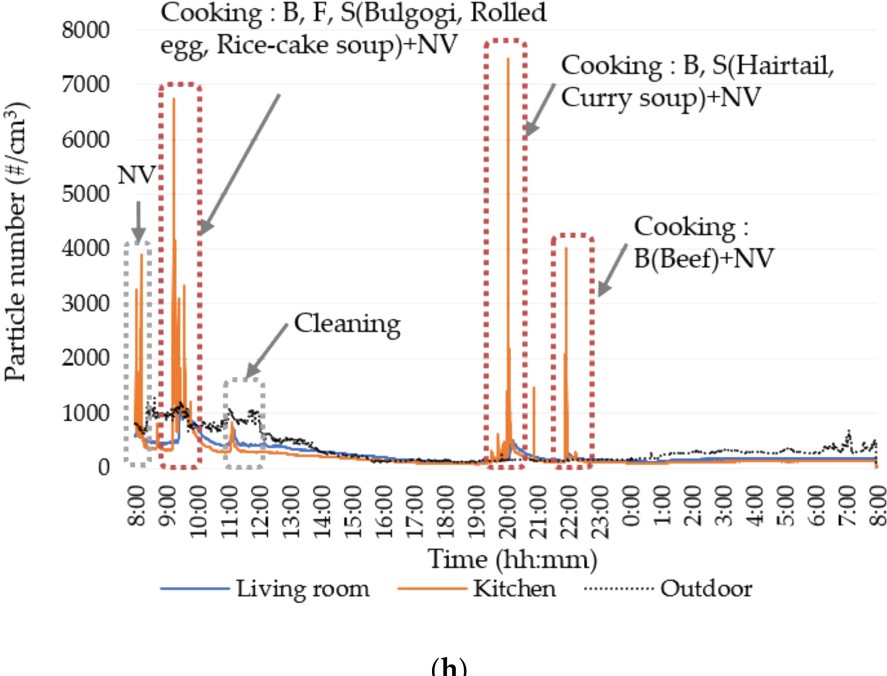

(**h**)

**Figure A1.** *Cont.*

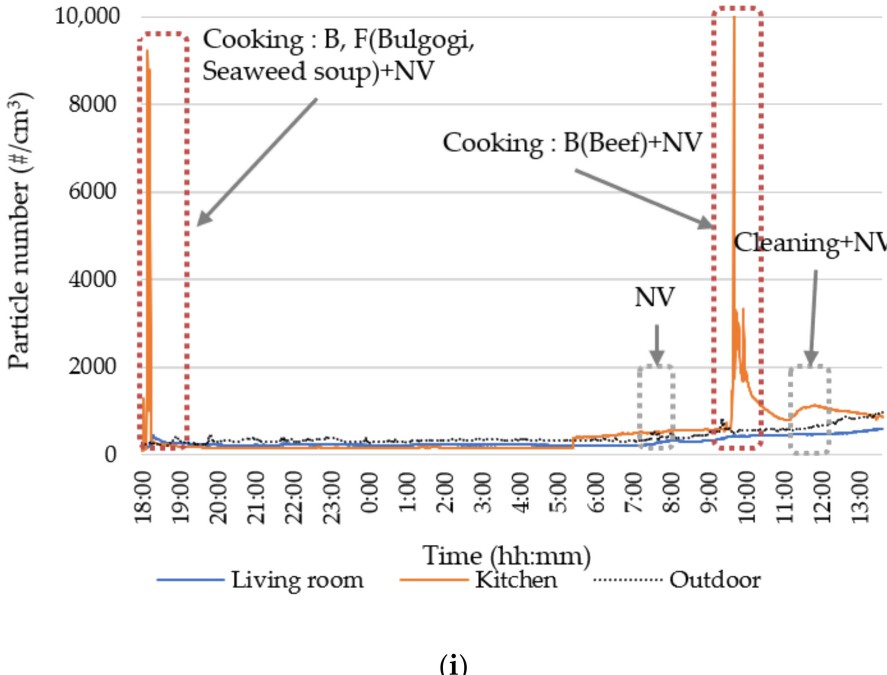

(**i**)

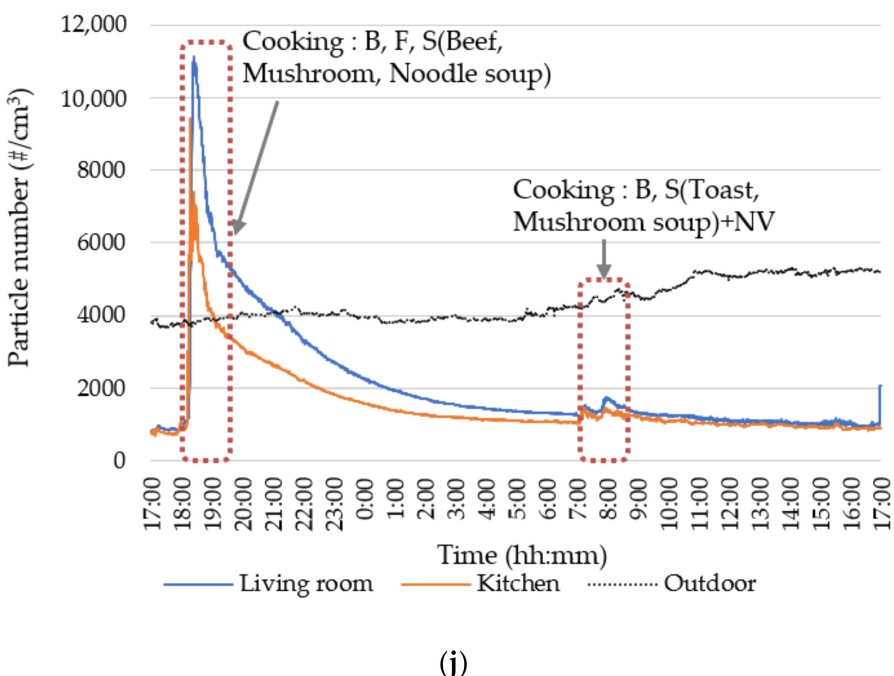

(**j**)

**Figure A1.** Particle concentration of the kitchen and living room and occupant activities. (**a**) Particle concentration of H1; (**b**) Particle concentration of H2; (**c**) Particle concentration of H3; (**d**) Particle concentration of H4; (**e**) Particle concentration of H5; (**f**) Particle concentration of H6; (**g**) Particle concentration of H7; (**h**) Particle concentration of H8; (**i**) Particle concentration of H9; (**j**) Particle concentration of H10.

## Appendix B. Particle Concentration of Outdoor Air

**Table A1.** Description of target buildings.

| Particle Size (µm) | | H1 | H2 | H3 | H4 | H5 | H6 | H7 | H8 | H9 | H10 |
|---|---|---|---|---|---|---|---|---|---|---|---|
| 24 h | 0.3–0.5 | 346.18 | 785.93 | 1347.29 | 1635.36 | 2894.52 | 2836.77 | 257.76 | 306.75 | 484.29 | 3309.72 |
| | 0.5–0.7 | 44.62 | 99.43 | 150.34 | 206.65 | 440.73 | 522.52 | 33.21 | 35.85 | 64.69 | 900.44 |
| | 0.7–1.0 | 10.07 | 21.33 | 24.15 | 36.52 | 62.53 | 67.19 | 8.70 | 8.95 | 13.30 | 142.59 |
| | 1.0–2.5 | 2.39 | 4.40 | 3.61 | 5.16 | 8.85 | 8.64 | 2.09 | 1.99 | 2.21 | 12.74 |
| | 2.5–5.0 | 1.21 | 1.26 | 0.66 | 0.70 | 1.23 | 1.00 | 0.43 | 0.46 | 0.53 | 1.69 |
| | 5.0–10 | 0.32 | 0.23 | 0.08 | 0.09 | 0.12 | 0.10 | 0.06 | 0.08 | 0.11 | 0.19 |
| | Total | 404.80 | 912.58 | 1526.12 | 1884.48 | 3407.98 | 3436.22 | 302.26 | 354.09 | 565.12 | 4367.38 |
| Cooking condition | 0.3–0.5 | 423.44 | 366.28 | 883.26 | 303.50 | 2250.74 | 1550.19 | 279.46 | 119.70 | 478.71 | 3022.09 |
| | 0.5–0.7 | 61.20 | 49.58 | 93.73 | 205.07 | 289.76 | 299.45 | 70.12 | 19.04 | 59.37 | 698.52 |
| | 0.7–1.0 | 15.04 | 12.61 | 18.20 | 29.07 | 42.72 | 48.19 | 27.08 | 5.41 | 13.40 | 15.04 |
| | 1.0–2.5 | 3.57 | 4.56 | 3.42 | 4.04 | 5.64 | 5.80 | 9.28 | 1.36 | 2.32 | 11.59 |
| | 2.5–5.0 | 1.16 | 2.74 | 0.72 | 0.64 | 0.75 | 0.59 | 1.09 | 0.34 | 0.66 | 2.16 |
| | 5.0–10 | 0.30 | 0.59 | 0.09 | 0.08 | 0.09 | 0.08 | 0.10 | 0.05 | 0.15 | 0.30 |
| | Total | 504.71 | 436.36 | 999.42 | 542.4 | 2589.7 | 1904.3 | 387.13 | 145.9 | 554.61 | 3749.7 |

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
