# Peer review of "Effect of Occupant Activity on Indoor Particle Concentrations in Korean Residential Buildings"

_sustainability, doi:10.3390/su12219201_

Round 1

Reviewer 1 Report

The manuscript analyses indoor and outdoor air concentrations in several apartments in South Korea. Even though it may seem interesting, the quality and content of the manuscript have to be improved. Please find attached some recommendations to do so:  

Introduction: The authors should highlight more negative effects of indoor pollutants on people's health and behaviour, since little information about this matter is provided. Likewise, positive effects of a good IAQ ambient should be included in the text, refering also to people's moody and behavioural aspects. Thus, a more extensive literature review have to be provided in the introduction: Are there similar studies? What do they analyse? What are their main conclusions? Are there other studies in the same area and what do they report? And so, what is the innovative approach of the present paper? Why is this research different from other studies?    

Methodology: In subsection 2.2, the specific indoor and outdoor particles measured must be included. What do the authors refer to when saying "particle concentrations"? (i.e., CO2, CO, etc.). In Table 1, it is suggested to include if there's cross natural ventilation in "Type of Ventilation". The authors should also specify the ventilation surfaces (m2) or the ventilation rates (ACH) related to both natural ventilation and mechanical ventilation of rangehoods. (The rangehoods rates may seem to be provided in Table 2, but no reference to the natural ventilation rate is made and these aspects have a noticeable influence on the conclusions the authors may report).  In subsection 2.4, do the I/O and L/K ratios have a limited or maximum recommended value? If not, how may the authors consider that IAQ is good or aceptable? Those variables should be compared to a reference value to report solid conclusions. What have other studies reported regarding these indexes? As authors analyse, indoor "particle concentrations" when compared to outdoor, the location of the houses must be provided (rural, urban areas... with low, medium, high outdoor concentrations...). i.e., when determining  I/O ratio, results will be equal when considering 500/1000 as 50/100. Perhaps the authors should include average outdoor values (Table 3 only shows indoor values). Likewise, another aspect that is key and the authors have not mentioned is the floor where the analysed house is (first floor, fifth floor...?). Figure 5 and the ones in the Appendix should be compared to maximum recommended "particle concentration" values.    

Results and conclusions: Once improving the manuscript taking into account the comments made, new conclusions and comparisons should be reported. The authors mentioned in the conclusion section that "recently built apartment houses have excellent airtightness". Maybe the authors should also include the construction date of the houses that have been monitored. Beside the measurement period limitation, what other future studies may the authors conduct?    

Author Response

We are grateful to the reviewers for a very thorough review and constructive comments. The comments are listed below with our responses. We have revised the manuscript based on reviewers’ comments. We hope that our responses and the changes in the revised manuscript are sufficient for the reviewers.

  1. Introduction: The authors should highlight more negative effects of indoor pollutants on people's health and behaviour, since little information about this matter is provided. Likewise, positive effects of a good IAQ ambient should be included in the text, refering also to people's moody and behavioural aspects. Thus, a more extensive literature review have to be provided in the introduction: Are there similar studies? What do they analyse? What are their main conclusions? Are there other studies in the same area and what do they report? And so, what is the innovative approach of the present paper? Why is this research different from other studies?

Response:

We agree with the opinion that the introduction part needs to be supplemented. We have added some statements to the introduction part by accepting the review’s comments.

-     We have added some explanations about negative effects of indoor PM on occupants. (line 31, 43-48)

-     We have added some statements about positive effect of a good IAQ including mental health of occupants. (line 54-58)

-     A more extensive literature review has been added. It also mentioned the conclusion of the study. (line 59-65)

-     We have added some statements to complement the novelty of this study. (line 74-76, 78)

  1. Methodology: In subsection 2.2, the specific indoor and outdoor particles measured must be included. What do the authors refer to when saying "particle concentrations"? (i.e., CO2, CO, etc.). In Table 1, it is suggested to include if there's cross natural ventilation in "Type of Ventilation". The authors should also specify the ventilation surfaces (m2) or the ventilation rates (ACH) related to both natural ventilation and mechanical ventilation of rangehoods. (The rangehoods rates may seem to be provided in Table 2, but no reference to the natural ventilation rate is made and these aspects have a noticeable influence on the conclusions the authors may report). In subsection 2.4, do the I/O and L/K ratios have a limited or maximum recommended value? If not, how may the authors consider that IAQ is good or aceptable? Those variables should be compared to a reference value to report solid conclusions. What have other studies reported regarding these indexes? As authors analyse, indoor "particle concentrations" when compared to outdoor, the location of the houses must be provided (rural, urban areas... with low, medium, high outdoor concentrations...). i.e., when determining  I/O ratio, results will be equal when considering 500/1000 as 50/100. Perhaps the authors should include average outdoor values (Table 3 only shows indoor values). Likewise, another aspect that is key and the authors have not mentioned is the floor where the analysed house is (first floor, fifth floor...?). Figure 5 and the ones in the Appendix should be compared to maximum recommended "particle concentration" values.

Response:

The reviewer gave us a lot of comments about the methodology in this manuscript and we agree with most of them. Considering the reviewer's comments, we have revised many parts.

-     We have added specific range. (line 109)

-     At the time of measurement, it was difficult to investigate the area of windows or the detailed ventilation type due to personal privacy. And the measurements were conducted in winter, so occupants didn't open all the windows and didn’t do cross ventilation. Therefore, single sided ventilation was performed for all generations. And when they operated the range hood, they opened the window and supplied make-up air for range hood. Therefore, it could be assumed that the airflow rate of natural ventilation is approximately the same as the amount of exhaust airflow rate in the range hood. I've added some explanations about this. (line 96-101, 169-171)

-     All the target buildings are located in the urban area. (line 86) To compensate for this, figure 1 was added. (line 88, 89, 105)

-     The I/O ratio may vary significantly depending on the region and residents' activities. I have added some literature to allow the reader to consider this. (line 139-146)

-     We also added some statement about L/K ratio. (line 151-154)

-     We added the average concentration of outside air in Appendix B.

-     The number of floors in the building is in Table 1. But it can be difficult for readers to understand. To make up for this, we added an additional explanation under the table 1. (line 103)

-     Although there is a concentration standard of PM2.5 or PM10, it is difficult to find the maximum recommended "particle concentration" of number concentration. To compensate for this, however, we added the measurement results of number concentration during cooking from the previous studies. And the measurement data were similar to the results of this study. (line 245-253)

  1. Results and conclusions: Once improving the manuscript taking into account the comments made, new conclusions and comparisons should be reported. The authors mentioned in the conclusion section that "recently built apartment houses have excellent airtightness". Maybe the authors should also include the construction date of the houses that have been monitored. Beside the measurement period limitation, what other future studies may the authors conduct?

Response:

We revised some of the conclusions and discussions after accepting the reviewers' opinions.

-     The built year of H1-H10 is added to Table 1.

-     The last sentence has been modified. (line 281-282)

Reviewer 2 Report

While I appreciate the authors examining occupant activity and its impact on indoor particle concentrations, I have few methodological queries that needs clarification. Specifically,

1) How did the authors choose the 10 apartments in South Korea? Is it random sampling? Can you please provide map indicating the location of these apartments?

2) I am not particularly keen on how the authors use the words "no clear relationship" and "significant effect" in the manuscript as they have not shown any statistical methods to prove this. I find this particularly unconvincing in the following statements:

a) Lines 144-145: "There was also no clear relationship between the ventilation frequency and I/O ratio." <- How did you prove this lack of clear relationship statistically?

b) Lines 157-158: "This appears to be because most of the large-size particles were generated when particles were generated by indoor occupant activities. " <- Can you show the proportion of the large size particles generated from indoor occupant activities?

c) Lines 166-168: "First, it appears 166 that the outdoor particle concentration did not have a significant influence on the indoor particle concentration in most cases." <-How did you prove this there is no significant influence statistically?

d) Lines 171-173: "During the measurement period, cleaning was performed in all the houses, but there was no significant change in indoor particle concentration before and after cleaning." <- How can you prove that there is no significant change before and after cleaning, statistically? there should always be a statistical theory underpinning the use of "statistical" change.

Author Response

We are grateful to the reviewers for a very thorough review and constructive comments. The comments are listed below with our responses. We have revised the manuscript based on reviewers’ comments. We hope that our responses and the changes in the revised manuscript are sufficient for the reviewers.

  1. How did the authors choose the 10 apartments in South Korea? Is it random sampling? Can you please provide map indicating the location of these apartments?

Response:

The 10 apartments were selected by residents of apartment houses in urban areas with the intent to participate. To describe this, figure 1 was added. (line 86-89)

  1. I am not particularly keen on how the authors use the words "no clear relationship" and "significant effect" in the manuscript as they have not shown any statistical methods to prove this. I find this particularly unconvincing in the following statements:
  2. a) Lines 144-145: "There was also no clear relationship between the ventilation frequency and I/O ratio." <- How did you prove this lack of clear relationship statistically?

  1. b) Lines 157-158: "This appears to be because most of the large-size particles were generated when particles were generated by indoor occupant activities. " <- Can you show the proportion of the large size particles generated from indoor occupant activities?

  1. c) Lines 166-168: "First, it appears 166 that the outdoor particle concentration did not have a significant influence on the indoor particle concentration in most cases." <-How did you prove this there is no significant influence statistically?

  1. d) Lines 171-173: "During the measurement period, cleaning was performed in all the houses, but there was no significant change in indoor particle concentration before and after cleaning." <- How can you prove that there is no significant change before and after cleaning, statistically? there should always be a statistical theory underpinning the use of "statistical" change.

Response:

We agree with the opinion that the introduction part needs to be supplemented. We have revised original manuscript by accepting the review’s comments.

a)   We agree with the reviewer that statistical analysis is required. However, with the data measured in this study, the number of samples is small, making it difficult to perform statistical analysis Nevertheless, we added some explanations to make up for this. First, table 3 was added to present quantitative data of I/O ratio. T Table 3 shows the I/O ratio of 24 hours average, ventilation and cooking condition. And we added some statements to explain table 3. (line 186-195)

b)  We think this sentence may cause misunderstanding to the reader. Therefore, we deleted this sentence.

c)   This statement overlaps the description in Chapter 3.1. To avoid this, we modified this sentence. (line 217-219)

d)  We agree with the opinion that this statement needs to be supplemented. Table 4 has been added to compensate for this. And we added a description for table 4. (line 223-229)

Round 2

Reviewer 1 Report

The reviewer thanks the authors for addressing the suggestions and comments made. The reviewer belives the manuscript has clearly improved from the previous state and is now acceptable for publication

Author Response

We are grateful to the reviewers for a very thorough review and constructive comments.

Reviewer 2 Report

I am afraid that the limited number of the sample locations may not be reflective of the totality of situation in the bigger population, thus would render conclusions moot.

Author Response

We are grateful to the reviewers for a very thorough review and constructive comments. We have revised the manuscript based on reviewers’ comments. We hope that our responses and the changes in the revised manuscript are sufficient for the reviewers.

- We agree with the opinion that the Some evidence about the representativeness is necessary. We have added some explanations by accepting the review’s comments. (line 87-90)

- Although there are not many samples in this study, they are not that small compared to other previous studies. To compensate for this, we added the number of buildings that have been investigated in other studies. (line 61, 64, 144-146, 149, 151)

- As the reviewer pointed out, there are some limitations to this study. To explain this, several sentences were added to the conclusion and abstract. (line 22-23, 288-292)

Round 3

Reviewer 2 Report

The respective comment/s were addressed sufficiently.